# Industrial Packaging Performance Indicator Using a Group Multicriteria Approach: An Automaker Reverse Operations Case

**Marcelo Miguel da Cruz** [1] , **Rodrigo Goyannes Gusmão Caiado** [2] **and Renan Silva Santos** [1,2,*]

1 LATEC—Laboratory of Technology, Business Management and Environment, Federal Fluminense University (UFF), Niterói, Rio de Janeiro 24000-000, Brazil
2 Industrial Engineering Department, Pontifical Catholic University of Rio de Janeiro (PUC-Rio), Rio de Janeiro 22541-041, Brazil
* Correspondence: renansilvasantos@esp.puc-rio.br

**Abstract:** *Background*: Due to the growing integration between the various logistics entities and other internal operations, packaging management in the automotive industry is becoming increasingly important from the strategic point of view of the logistics operations of automakers. Performance evaluation of reverse operations is also necessary for managers to know their efficiency, avoid unnecessary resource use and promote circular thinking, enabling more sustainable supply chains. *Methods*: This research proposes a group decision-making (GDM) approach to evaluate packaging performance in automakers to assist return activities in developing countries. The reverse flow in an automaker was mapped, and by combining literature and empirical views of a packaging engineering team of a Brazilian company, a multicriteria indicator for performance evaluation of packaging was elaborated. It was prioritized through the analytic hierarchy process (AHP)-GDM method, combining judgments to establish a structured technical consensus. *Results*: It was possible to integrate multiple views of packaging engineering specialists within the same company to know which packaging deserves greater attention from managers when implementing reverse operations from a circular perspective. *Conclusions*: To demonstrate applicability, this composite indicator also aims to be a quick application approach, considering the restricted time and availability of the specialists in their daily routines.

**Keywords:** packaging development; AHP-GDM; automakers; automotive industry; reverse logistics

---

## 1. Introduction

Although packaging can be seen as an aggregating value to products in retail (such as supermarkets, groceries, and stores), it is also one of the determining factors in increasing efficiency for supply chain management in automotive industries, including vehicle manufacturing. The authors of [1] cite that packaging can be defined as the integrated system of materials and types of equipment with which one takes goods and services to the hands of the end consumer using distribution channels. In order to present a complete definition, the authors state that in a more abstract sense, the packaging would be the set of arts, sciences, and techniques used in the preparation of the products to create the best conditions for transport, storage, distribution, sales and consumption or, alternatively, the means to ensure delivery of a product in a reasonable condition with the minimum total cost. According to Ref. [2], besides the main functions, packaging affects the supply chain in several ways, given its interactions with logistics, manufacturing, and other systems, and the environmental performance regarding waste and transport fill rates.

The automotive industry supply chain involves automakers, suppliers, retailers, and the final consumer. A single delay generated by any member can cause production stoppages and, consequently, high economic losses [3]. Regarding the types of packaging used





to transport automotive parts, the authors of Ref. [4] find that returnable plastic or metal packaging (including pallets, containers, and racks) is used by most companies. However, they are not limited to these types of materials. In the case of the automotive industry, the packaging used to transport the assembly parts of the vehicles is chosen after fulfilling promises related to financial, technical, and environmental factors (to obtain better performance/results for the assemblers).

The choice of the best type of packaging to be adopted for each application is previously analyzed by specialists, usually engineers and/or packaging analysts, who have the task of optimizing the volume occupied by each part inside the packaging, thus rationalizing all the processes in which the packaging goes through in its use flow from the parts supplier to the automakers. However, when trying to optimize the use of these packages, these specialists face other elements related to implementing a given package in the flow of parts supply. Among these elements is the definition of the type of packaging to be used in this flow. Thus, when choosing the best type of packaging, whether returnable, disposable or reusable, issues related to sustainability, cost−benefit, handling, and durability, among others, arise as essential points for a better evaluation of packaging previously defined by a specialist.

In general, most automakers choose to adopt or at least prioritize the use of returnable packaging in their operations simply because they understand that the use of this type of packaging in the flow will allow a significant reduction of the adverse financial effects caused by the use of disposable materials and elements, considered as mainly responsible for indirectly burdening the unit cost of vehicles produced by the plant itself; i.e., there is a cost correlated to the disposal of materials that impacts the profitability of companies.

As highlighted by [5], returnable items play an essential role in modern logistics because their use in the transportation of products along the supply chain can bring several benefits, such as waste reduction, better product protection and safety, more efficient handling, better opportunities for outsourcing, combining and standardization, and reduction of $CO_2$ emissions throughout the packaging life cycle. Returnable packaging management is an important issue in automotive parts logistics, especially for the automotive industry, which faces cost reduction pressure due to increased competition and lower profit margins [4].

Thus, practical and applicable solutions such as the one presented in the study by [2], in which several criteria were taken into account in the construction of a decision-making model about the management of industrial packaging waste used in the supply chain of an automaker (both in terms of $CO_2$ emissions and the operating costs involved from a sustainable perspective of its supply chain), indicate that the development of a management model and packaging performance evaluation can help to better inform management decisions.

As the authors of [6] mention in their study, the set of ESG (environmental, social, and governance) ratings and benchmarking that are currently available provide a standard by which business performance can be monitored and compared by external stakeholders. In other words, jointly involving different stakeholders and dimensions (as is the case with the ESG perspective) may generate even better standards when compared to the current ones, establishing a way to compare and/or monitor the evolution of organizational performance against itself (when comparing the result from one year to another) and against its competitors (when a better strategic positioning is desired), without leaving aside sustainability aspects.

These aspects align with the increased interest observed in recent years regarding the circular economy (CE). CE models and practices provide a framework opposed to the more traditional "take-make-dispose" approach, which has negatively impacted the environment over time [7]. Thus, CE practices have been studied in several application scenarios among both companies, governments, and other stakeholders [8–13]. Even aspects such as how ERPs can be used to leverage and enhance CE processes have been studied [14]. In this context, the role of packaging in facilitating aspects of CE regarding

sustainable practices that affect both production and consumption, such as reverse logistics, is of utmost importance [15].

Although this trend is real and feasible from the sustainability point of view (at least from the environmental point of view), we still have to ask some questions on this subject, such as: "Is this packaging the best and most appropriate to be used in the logistics operations of a given company?"; "What are the justifications for technicians choosing to use packaging X over packaging Y?"; "Is it possible to quantify in the form of an indicator the parameters and settings of a packaging product, thus evaluating its performance?". With these questions in mind, packaging development experts could answer all these questions with a straightforward answer: "It depends, each case is a separate case!". Thus, faced with this gap related to vagueness and subjectivity linked to the personal choice of the expert on what type of packaging would be best in each case, an opportunity arises to create more practical, rational, and optimizable means and/or methods that facilitate decision-making by experts (engineers and analysts) in this area. The CE aspects of packaging can also be considered in a structured decision support system [16].

The authors of [2] in their study developed a scorecard model that allowed the judgment and performance evaluation of the packaging in different parts of a supply chain from the perspective of environmental and economic sustainability. Similarly, there is an opportunity to develop a model capable of combining specialists' perceptions in bag engineering under a hybrid approach that is simple to apply within an organization. After all, much of the experts' time is spent searching for ways in which they can justify the "how" and "why" they chose to select a given packaging for a given component (assembly part). Thus, this study tries to solve these doubts and research questions and establish the rational and well-founded use of common sense by the specialists (current situation). To this end, it is necessary to create a solution related to the theme, as long as it can benefit those responsible for the packaging development (specialists) and the operational results of the assemblers that adopt it in general.

Therefore, this research aims at proposing a group decision-making (GDM) approach to quantify (in the form of a ranking) the choices and definitions related to the packaging and transport of parts in a company in the automotive sector (vehicle assembler, located in the south of the State of Rio de Janeiro, which for confidentiality reasons will be called "XPTO AUTOMOTIVE" throughout the study), considering the most important criteria (priority) from the perspective of a group of packaging specialists. In summary, it is expected that the proposed research approach will be a means to simplify the decision-making by packaging development specialists about which packaging model to use, to become more assertive, and to assist in the specialists' decisions (daily routines) about which would be the best packaging to be used for a specific part of a vehicle, via the application of a multicriteria decision-making model (in group) objective and clear for the users. The study proposes a framework based on four steps, the CEPA (contextualize−explore−prioritize−aggregate) framework. The novelty of this research, besides the CEPA framework, lies in the fact that it is a case study for application of this proposed framework in an automaker, with real experts forming the panels.

Because the high costs of improvement involved in I4.0 and sustainability are the most significant economic challenge presented in the literature, the investment needed and the capital expenditures underlying I4.0 technologies are quite intense, especially for manufacturing located in the context of emerging economy companies [17].

However, it also is worth considering that the proposed application can be quickly implemented through a simple Microsoft Excel spreadsheet, reducing the costs of implementing the model (although acquisition of a specific software license would require training those involved, as well as purchasing this license, generating extra costs for the organization). Thus, the solution presented in the present work becomes viable to be adopted by companies.

## 2. Literature Review

### 2.1. Brazilian Automotive Industry

The automotive sector is a significant driver of the national and global economy and fosters innovation greatly due to the industry's high competitiveness and its substantial share in overall trade [18]. It is considered a sector that is a technological vanguard, with intense competition among the companies involved, and one of the industrial sectors that employs the most in Brazil. With the entry of numerous automakers into the Brazilian market, and due to the high competitiveness of the automotive sector, obtaining a competitive advantage by improving logistics processes has become a determining factor for the survival and success of these companies. Reducing costs and minimizing the environmental impact of packaging is a critical element of business operations in today's world [18].

### 2.2. Packaging in the Automotive Industry

The use of packaging in the automotive industry arises from the need to transit parts between parts suppliers and vehicle assemblers through a continuous supply, transport, storage, consumption, and return of parts. It can be understood that packaging development and solutions need to facilitate or optimize inter- and intra-process movements so that the chances of damage (and/or loss) to the transported products are reduced and facilitate the management of these assets (packaging).

Packaging is part of the companies' SCM (supply chain management) process, where, according to [3], due to the complexity of parts and components in transit operations in the automotive industry, it is more than necessary to achieve success in supply chain management, mainly by reducing logistics costs and integrating those involved. With regard to the materials most frequently used in the manufacture of packaging, the following stand out: cardboard, wood, metal, Styrofoam, EVA, plastic, and corrugated plastic. These materials are used individually and/or in combination, forming new packaging concepts. In general, experts in automotive industry packaging tend to segment/classify the packaging as follows: the type (disposable and reusable), the materials primarily used in their composition (wood, cardboard, plastic, and metal), and as to their specificity (standard or specific). According to [4], when comparing returnable packaging to disposable packaging, the first can reduce the total amount of packaging needed due to its longer shelf life, which is more ecological and better for sustainability. This fact corroborates the context in which companies in the automotive sector operate, of intense competition and the search for lower costs for their products.

### 2.3. Packaging Development and Performance in the Automotive Industry

Once aware of the characteristics attributed to each of the packaging models available, some packaging specialists end up opting for excessive zeal, emphasizing the safety and quality assurance of the parts being transported in the packaging, even if the level of risk is historically low. On the other hand, other packaging specialists value the elaboration of packaging that reinforces the concept of minimalism (such as "less is more") in its creative essence, i.e., they try to use as few accessories and/or additional elements as possible to protect and accommodate the parts inside the packaging, which may be the cause of increased risk of part quality failure and losses. Additionally, as in several other industries, there has been an increased call for more sustainable practices in packaging, which can enable a cleaner approach in both production and consumption stages [7,15], increasing complexity of decisions regarding packaging. Thus, it can be concluded that there is no technical−practical consensus among these specialists, which gives the decision-making about the choice of a given packaging a more subjective bias. Each specialist has an opinion about the packaging solution to be implemented for a given situation. Thus, studies such as that of [19] justify the use of clear performance indicators for packaging development in the automotive industry.

### 2.4. AHP-GDM

The AHP method, developed by [20], is one of the most used multicriteria methods, which could be related to its practical structure and fast construction. Its development starts from an intuitive idea and/or problem, structured hierarchically and logically, making it possible to judge the available criteria and/or options objectively. Another essential feature of the AHP is that it explicitly evaluates the consistency of the judgments elicited by decision-makers to incorporate their preferences [21].

Researchers in many fields have used AHP-GDM (analytic hierarchy process—group decision-making), in which decision-making is performed through the consensus of a group, following the pre-established steps in the AHP itself. AHP is one of the most popular decision support tools in GDM involving multiple actors, scenarios, and criteria [22]. GDM refers to processes where the decision made is no longer associated with a single person, but with the entire group, since many group members contributed to its adoption [23]. Group decision-making involves the weighted aggregation of different individual preferences to obtain a single collective preference [24]. Various factors determine successful decisions, including the ability of group members to communicate effectively and reach efficient decisions [23].

Some studies propose a composite indicator for reverse logistics performance evaluation in civil construction by using a simple AHP procedure [25]. However, there are no studies combining AHP-GDM within a framework to develop indicators for assessing sustainable packaging in the automotive industry and from a circular economy perspective.

### 3. Application of the Method

This section describes the methods used to develop this research to detail the application of the multicriteria group decision-making method (AHP-GDM) to model and propose a performance indicator capable of assisting in the definition of packaging used in the supply of parts in a vehicle manufacturer.

### 3.1. Background

The object of analysis is a multinational company in the automotive sector, which produces passenger cars sold in the Brazilian and international markets (MERCOSUR countries), with a manufacturing plant in the south of the State of Rio de Janeiro. However, for confidentiality reasons, the name of this assembler was changed to XPTO AUTOMOTIVE. In general, the projects concerning new vehicles launched by this company are originated from its head office; that is, the projects are created in another region of the world, to then be adapted, or "tropicalized" (as it is usually said in the language of the sector) for manufacture in Brazil, as is done by most of the other automakers established in the country.

For the production of vehicles in its plant, the automaker acquires its assembly parts both from local suppliers (in Brazil) and from imported suppliers (in this case, mostly from suppliers located on the American, European, and Asian continents). However, although there is the purchase of parts via imports, it can be said that in terms of percentage, there is a greater representativeness of assembly components from Brazilian suppliers (local) in this mix. However, among the several processes and phases that involve the internal logistic flow of materials—which involve stages from the purchase to the consumption of parts at the line edge by operators responsible for vehicle assemblies—there is the process of defining packaging for transporting parts between the supplier and the production plant, a process that may involve several different areas/sectors and/or some pre-established organizational aspects (aligned with the company's strategic objectives).

Thus, in an attempt to define and/or choose the most appropriate packaging for each part transportation logistics process (between suppliers and manufacturing plant) or for each of the vehicle assembly processes, there is a packaging engineering team, which is dedicated not only to deal with the definition of packaging, but also to establish the parameters related to the supply of parts by suppliers based on certain criteria, which so

far can be considered as subjective criteria in the light of the professionals and specialists who work in this area (packaging engineering sector).

Thus, given the above, the nine professionals who are involved in the process of packaging development at the assembler, from the management members to the staff members of the packaging engineering team, are hierarchically linked to the supply chain management sector of XPTO AUTO-MOTIVE, and are listed and organized according to Figure 1 and Tables 1 and 2 below, which illustrate and detail respectively: the hierarchical structure (organization chart), the panel of specialists (background and responsibilities), and the technical-co-professional experience time of the packaging engineering team of the assembler that is the object of study.

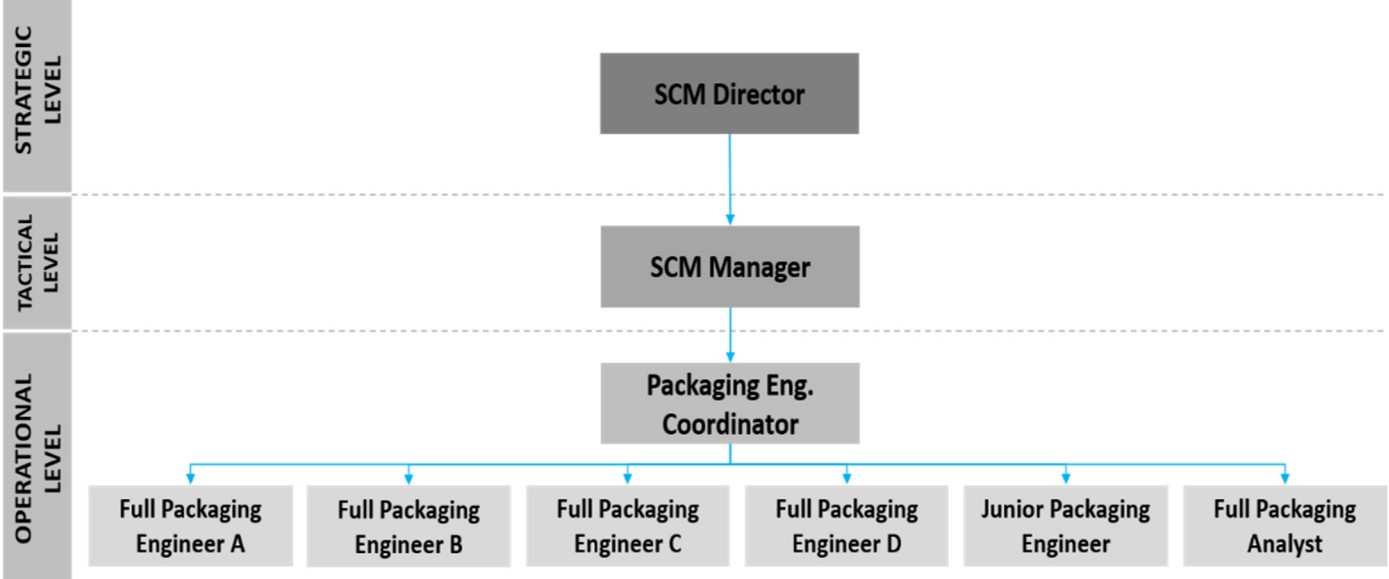

**Figure 1.** Hierarchical structure (organization chart) of the packaging engineering team at XPTO AUTOMOTIVE.

**Table 1.** Expert panel (qualifications of the respondent sample).

| Function/Job Title | Background | Responsibilities/Activities |
|---|---|---|
| **SCM Director** | Professional with experience in various areas of the automotive industry. He has international experiences within his sector of activity (SCM), although a few years focused on the direction of the packaging development engineering team. | Acts as a link between the factory and the company's other plants worldwide. Responsible for steering meetings that involve flow changes and strategic decisions for the organization as a whole. He is very present in decisions that involve new packaging projects. |
| **SCM Manager** | He has experience in freight scheduling and negotiating transport contracts with carriers and freight forwarders at an international level. He has also worked in studies related to foreign trade (import/export) in several types of models. He has knowledge related to customs and port logistics operations. | Responsible for managing the packaging development team and two other teams related to logistics, one for scheduling and optimizing transportation (inbound logistics) and the other for international logistics (focused on the import and export of materials). |
| **Packaging Engineering Coordinator** | Professional with experience in transportation scheduling and the area of international freight forwarding, cargo clearance, and customs. | Acts as a link between top management and the packaging engineering team (staff), controlling work demands and piloting current and future projects related to packaging development. |

Table 1. *Cont.*

| Function/Job Title | Background | Responsibilities/Activities |
|---|---|---|
| **Full Packaging Engineer A** | He has expertise in developing packaging for metal parts and the external housing of vehicles. He has received international awards related to packaging development within the company. | Responsible for developing packaging for the transportation of metal parts, both for internal and external flow, in addition to being responsible for the maintenance control of all packaging circulating in the plant. |
| **Full Packaging Engineer B** | He has extensive experience in flow and internal logistics projects (balancing supply and line edge preparation). | Sole direct responsibility for packaging development of the international flow (imported parts), dealing with suppliers and other company plants worldwide. |
| **Full Packaging Engineer C** | He has previous experience in the development of specific packaging for metal parts in his former company, and has a lot of know-how in project management with a focus on cost management. | Responsible for the development of packaging in the local flow of suppliers, focused on interior trim parts of the vehicle; also performs the collation and internal analysis of costs related to packaging development (new projects or flow changes). |
| **Full Packaging Engineer D** | He has a vast background in the area of development of packaging for automotive parts, being the most experienced professional in the sector when it comes to the development of packaging, has a very analytical and broad vision of the entire development process of new projects, and is always consulted by the other team members about making decisions related to the theme. | Responsible for the analysis/study of new logistics projects, reporting directly to the management and direction of the sector. |
| **Junior Packaging Engineer** | He has experience in transportation flow analysis, scheduling, and freight forwarding, with recent experience in packaging development. | Responsible for the development of packaging related to the flow of parts for the final assembly of the vehicles and performing the studies related to freight costs and cubage improvement of the packaging developed by the rest of the team. |
| **Full Packaging Analyst** | He has previous experience in developing packaging for automotive parts and the production line supply flow at other assembly plants. | Responsible for the development of packaging for the assembly parts of the engines produced in the plant; also acts as the team's quality manager, responding to possible problems related to process failures involving packaging. |

Table 2. Professional experience of the team members.

| Function/Job Title | Industry in General | Length of Experience Automobile Industry | Packaging Development |
|---|---|---|---|
| **SCM Director** | 19 years | 19 years | 2 years |
| **SCM Manager** | 10 years | 4 years | 3 years |
| **Packaging Eng. Coord.** | 18 years | 18 years | 2 years |
| **Full Packaging Eng. A** | 9 years | 9 years | 5 years |
| **Full Packaging Eng. B** | 5 years | 3.5 years | 1.5 years |
| **Full Packaging Eng. C** | 10 years | 10 years | 7 years |
| **Full Packaging Eng. D** | 10 years | 10 years | 6 years |
| **Junior Packaging Eng.** | 10 years | 8 years | 8 years |
| **Full Packaging Analyst** | 16.5 years | 14.5 years | 12.5 years |

Analyzing the background of the professionals, one can see that it is a team of professionals with a lot of experience in their area of work and that have an interesting background to contribute to the success of this research.

*3.2. Research Method Application Steps*

The steps taken to apply the research method are illustrated in Figure 2. These steps form a framework which can be defined by the acronym CEPA (contextualize−explore−prioritize−aggregate), which summarizes the four main processes encompassed in each of the steps.

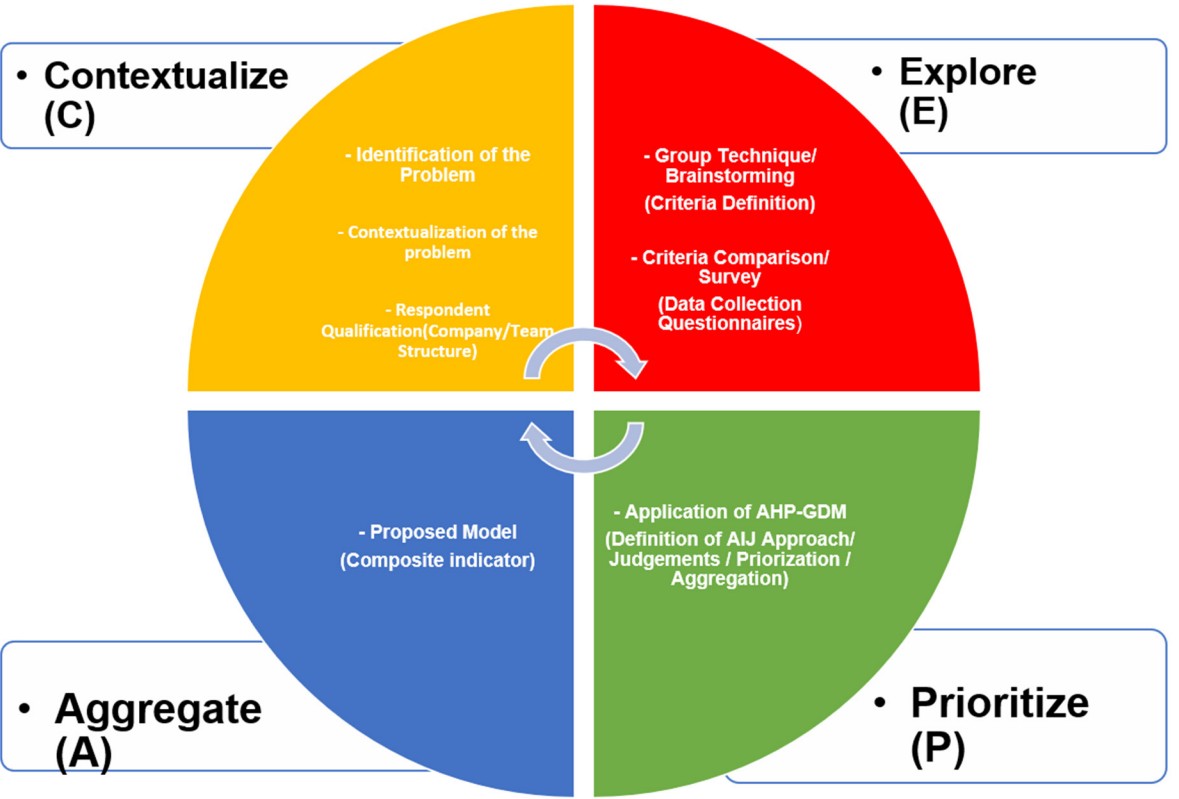

**Figure 2.** Research method application steps.

After applying the method outlined in Figure 2, conclusions, implications, considerations, and a research agenda can be defined.

In this work, the CEPA framework is used to guide the process of developing a composite indicator, from conception to the indicator itself. In the prioritize step, AHP-GDM was used as the tool to elicit priorities among the criteria that would compose the indicator, while allowing us to deal with possible inconsistencies in expert judgements.

*3.3. Current Status*

Given the contextualization exposed in Section 3.1, a problem situation comes to light that is part of the daily life of professionals in the area of development of industrial packaging of this assembler, which is related to current decision-making regarding the choice and definition of unique packaging for the transport of a given assembly part. Until then, these professionals are demanded to perform and execute continuous actions to reduce logistics and operational costs, because the packaging developed and defined by them goes through almost the entire production process of a vehicle and is considered a piece of equipment that directly impacts the operations.

However, when these specialists need to make a decision (define a packaging for a given part), they do not have at hand clear prioritization parameters aligned with what their managers want to improve performance indicators linked to their respective areas. In other words, the specialists end up assuming all the burden of their decision-making because there is no clear way to argue and justify their decisions. Thus, there is no objective and direct way to align expectations between their decisions and the opinions of their

leaders (managers of the packaging development area), who are able to base them on the decision-making (opting for the choice of one packaging instead of another for the transportation of parts). Through these observations, we identified a gap between the different arguments, which until then had been decisions purely based on subjective criteria (based on: "I think this packaging is better because it does not use disposable materials"; "I think this other packaging is better because it does not require equipment to move it"; "I think this one has a better use of cubic footage transported", and so each professional defines what he thinks is best for himself.

In view of this, there is a need to develop a method that is capable of aligning, in a grounded and standardized way, the decision-making perspectives of the specialists and the expectations of the area managers. Nevertheless, the fact that the process of approving the use of a given package in the process requires the approvals of several stakeholders linked to other areas of the company (professionals in the areas of quality, work safety, ergonomics, process engineering, transportation, operational logistics, costs, etc.). This makes the problem one even more necessary to solve, since this definition process is not clear and standardized in an explicit way so that there can be a discussion or a more substantiated debate between the parties, treating it similarly to other manufacturing processes inherent to the production of a vehicle, where there is a work standard to be respected.

### 3.4. Modeling (Packaging Performance Indicator)

This topic details the application of the multicriteria decision-making method (in group) adopted to propose a packaging performance evaluation indicator model, which is able to assist the decision-making of packaging development engineering specialists, about the definitions of packaging used in the logistics areas of the automotive industry, as well as the intermediate and/or complementary steps used for the elaboration of the model to be proposed.

### 3.4.1. Group Technique for Defining Criteria (Brainstorming)

To define the criteria, brainstorming was applied, which is a group dynamics technique commonly applied in the organizational environment to help explore the creative potential of a group or an individual towards one or more previously determined objectives in order to find new ideas, new concepts, and/or new solutions. As recommended by the authors of [26] in their study, this proposes the application of a group panel led by a facilitator who is able to understand the discussions about the new technology.

Thus, a brainstorming session was held, lasting 30 min, with the participation of the six members that make up the staff of the packaging engineering team of the company under study; i.e., only the members responsible for the actual development of packaging (analysts and engineers), because they are professionals who are in direct contact with both the teams linked to the manufacturing/production area and with the professionals linked to management, having to hear the opinions and interests of both parties.

In this brainstorming session, the criteria in Table 3 below were listed.

**Table 3.** List of criteria listed in the brainstorming session.

| Criteria | | | |
|---|---|---|---|
| volume | quality | development time | operational |
| cost | part quality | performance | locking system |
| acquisition cost | transport costs | engineering performance | use of moving parts |
| packaging type | disposable costs | maintenance costs | storage area costs |
| handling | inventory discrepancies | inventory losses | load and loading limitations |
| ergonomics | package durability | packaging management | frequency (packaging turnover) |
| safety | time | management | depreciation |

By analyzing this list, it was agreed among the experts present at the session that it would be interesting to limit the number of criteria to simplify the decision-making process. Thus, by consensus, it was possible to extract the six priority criteria (constructs) for packaging development in the light of these experts. The criteria chosen as priorities through the session were the following:

- Costs: Refers to the costs involved in the development and the costs incurred during the entire life cycle of these packagings. According to the specialists, the choice of this criterion was made due to its representativeness in the impacts related to the operational profitability of the plant (both in terms of CAPEX and OPEX);
- Quality: This represents the guarantee that the defined packaging protects the quality of the parts transported inside. Its importance is linked to the desire of experts to reduce the number of occurrences of quality breakdown (damage to parts) in which the packaging is considered the root cause;
- Safety: Refers to the definition that the packaging should respect the ergonomics and work safety rules/standards of the company and the local legislation, minimizing the risk of accidents. The specialists chose this criterion in order to avoid and minimize the impacts related to damaged packages during internal movements and to the number of accidents associated with each packaging model, as well as to help follow the internal ergonomic rules;
- Time: Refers to the prioritization of the use of packagings that have reduced times between the phases of a packaging project, which are: prototype, manufacturing in batches, and delivery to the plant. In order to reduce the total time interval that covers from conceptualization to availability and use of the packages in the flow (prototype, manufacturing in batches, and delivery), the specialist team judged that this criterion would bring benefits related to agility and compliance with the deadlines set by the plant regarding the start of the supply of a given part, especially during the launch of a new vehicle or in cases where modernization of a given vehicle model already existing in its production line occurred;
- Engineering Performance: Considers the use of optimizable factors related to the packaging construction, such as factors related to packaging with optimized cubic meters, and the increase of the valuable packaging life (durability)—prioritizing the use of returnable elements. This criterion considers the efficiency of the cubic footage occupied/cubic footage available (rate of occupancy of the packaging), the useful life of the packaging (in years), and the type of packaging (disposable or returnable);
- Management: This question is related to using packaging that allows easy operational management (minimum use of accessories) and offers future opportunities for flexibility regarding the use in other flows.

Thus, given the selection of these 6 (six) criteria and having as an objective the creation of a Packaging Performance Indicator, it was possible to arrive at the configuration of the AHP hierarchical decision tree for modeling the problem at hand, shown in Figure 3 below.

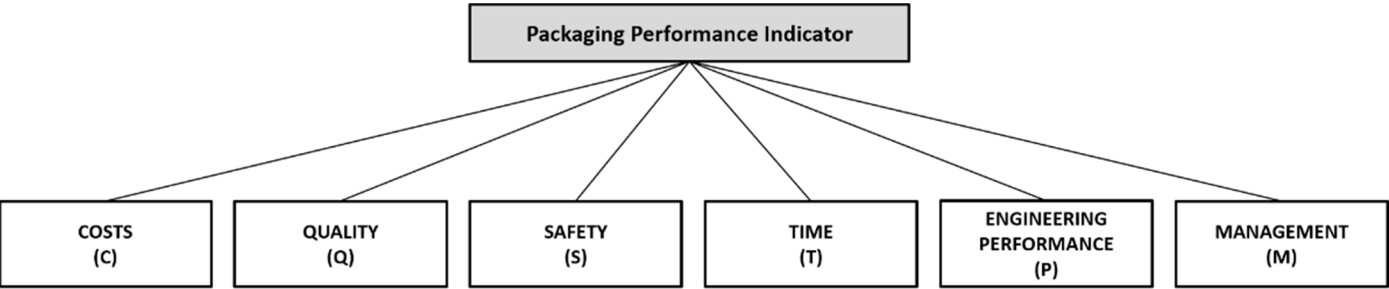

**Figure 3.** Model hierarchy tree.

### 3.4.2. Survey Application (Criteria Comparison)

The comparison/judgment between the criteria was performed by filling out a questionnaire (survey) applied via Google Forms.

In this questionnaire, the 9 (nine) specialists (managers, engineers, and analysts) could individually assign the intensity of importance for each of the 6 (six) selected criteria. Thus, Saaty [27] proposed that for applications of the AHP method, the comparisons between pairs of criteria were based on their fundamental scale of absolute numbers, respecting a linear scale from 1 to 9, shown in Table 4 below.

**Table 4.** Fundamental absolute number scale—adapted from [28].

| Intensity | Definition | Explanation |
|---|---|---|
| 1 | Equal importance | Two activities contribute equally to the objective |
| 3 | Slight importance | Experience and judgment slightly favor one activity over another |
| 5 | Strong importance | Experience and judgment strongly favor one activity over another |
| 7 | Very strong importance | An activity is favored very strongly over another; its dominance demonstrated in practice |
| 9 | Absolute importance | The evidence favoring one activity over another is of the highest possible order of affirmation |

The existence of reciprocity in these comparisons between criteria is also considered so that aij = 1/aji.

### 3.4.3. Choice of Approach (AIJ × AIP)

In the case of the present research, where the decision was being made in groups, it is necessary to define a method or approach for grouping the data as a way to treat the information obtained. The group decision-making theory related to the AHP determines that there are two ways to treat the obtained data:

AIJ—aggregating individual judgment:

This applies to the case where decision-makers participate in the same group, with the same culture, same preferences, and same values. Combining these factors causes the attributed judgments to become a group consensus. In other words, the harmony between the judgments made individually by each decision-maker will enable the group's decision, thus generating a new individual decision that unifies all the decision-makers. The homogeneity found in the group's decision makes the geometric mean calculation of each of the values Aij satisfy the group's hierarchical reciprocity in each decision-maker's judgment matrixes.

AIP—aggregating individual priorities:

The AIP approach applies to cases where the decision-makers are from different companies, tend to act according to their preferences, make decisions individually, and may be from different countries. They do not have a certain degree of rapport among themselves to the point of defining a decision for the good of the group. More straightforward than the AIJ approach, the simple arithmetic mean of the decision matrix vectors satisfies the condition of synthesizing individual priorities but does not impede the application of the geometric mean if the moderator feels more comfortable applying it.

As the present study aimed to establish an integration between the opinions of packaging engineering specialists within the same company, the AHP group decision-making method (for AHP-GDM) was applied, preferably in the AIJ—aggregating individual judgment form, in order to prioritize a consensus between components (experts and managers) of the analyzed group.

### 3.4.4. Individual Responses Obtained (Criteria Comparison Matrix)

With the answers obtained through the survey, where the specialists could contribute with their vision/opinion about packaging development, considering the organization's strategic vision in question, the individual judgment matrixes were elaborated for each one of these professionals. See Tables 5–13 below.

**Table 5.** Judgment matrix (SCM director).

| Criteria | (C) | (Q) | (S) | (T) | (P) | (M) | Eigenvector | W (Weights) |
|---|---|---|---|---|---|---|---|---|
| (C) | 1.00000 | 0.14285 | 1.00000 | 3.00000 | 0.14285 | 0.20000 | 0.48009 | 0.05996 |
| (Q) | 7.00000 | 1.00000 | 1.00000 | 5.00000 | 3.00000 | 7.00000 | 3.00410 | 0.37519 |
| (S) | 1.00000 | 1.00000 | 1.00000 | 5.00000 | 1.00000 | 1.00000 | 1.30766 | 0.16332 |
| (T) | 0.33333 | 0.20000 | 0.20000 | 1.00000 | 0.14285 | 0.14285 | 0.25456 | 0.03179 |
| (P) | 7.00000 | 0.33333 | 1.00000 | 7.00000 | 1.00000 | 0.14285 | 1.15167 | 0.14383 |
| (M) | 5.00000 | 0.14285 | 1.00000 | 7.00000 | 7.00000 | 1.00000 | 1.80860 | 0.22588 |
| Σ (Sum) | 21.33333 | 2.81904 | 5.20000 | 28.00000 | 12.28571 | 9.48571 | 8.00669 | 1.00000 |

**Table 6.** Judgment matrix (SCM manager).

| Criteria | (C) | (Q) | (S) | (T) | (P) | (M) | Eigenvector | W (Weights) |
|---|---|---|---|---|---|---|---|---|
| (C) | 1.00000 | 0.20000 | 0.20000 | 7.00000 | 1.00000 | 7.00000 | 1.11869 | 0.11746 |
| (Q) | 5.00000 | 1.00000 | 0.20000 | 5.00000 | 7.00000 | 5.00000 | 2.36505 | 0.24833 |
| (S) | 5.00000 | 5.0000 | 1.00000 | 7.00000 | 7.00000 | 7.00000 | 4.52417 | 0.47505 |
| (T) | 0.14286 | 0.20000 | 0.14286 | 1.00000 | 0.20000 | 0.33333 | 0.25456 | 0.02672 |
| (P) | 1.00000 | 0.14286 | 0.14286 | 5.00000 | 1.00000 | 5.00000 | 0.89390 | 0.09386 |
| (M) | 0.14286 | 0.20000 | 0.14286 | 3.00000 | 0.20000 | 1.00000 | 0.36713 | 0.03855 |
| Σ (Sum) | 12.28571 | 6.74285 | 1.82857 | 28.00000 | 16.40000 | 25.33333 | 9.52350 | 1.00000 |

**Table 7.** Judgment matrix (packaging engineering coordinator).

| Criteria | (C) | (Q) | (S) | (T) | (P) | (M) | Eigenvector | W (Weights) |
|---|---|---|---|---|---|---|---|---|
| (C) | 1.00000 | 0.20000 | 0.11111 | 0.11111 | 0.11111 | 0.11111 | 0.17674 | 0.01570 |
| (Q) | 5.00000 | 1.00000 | 0.20000 | 0.14286 | 0.11111 | 0.11111 | 0.34759 | 0.03087 |
| (S) | 9.00000 | 5.00000 | 1.00000 | 0.20000 | 0.11111 | 0.11111 | 0.69336 | 0.06158 |
| (T) | 9.00000 | 7.00000 | 5.00000 | 1.00000 | 0.20000 | 0.11111 | 1.38309 | 0.12285 |
| (P) | 9.00000 | 9.00000 | 9.00000 | 5.00000 | 1.00000 | 0.20000 | 3.00000 | 0.26646 |
| (M) | 9.00000 | 9.00000 | 9.00000 | 9.00000 | 5.00000 | 1.00000 | 5.65792 | 0.50254 |
| Σ (Sum) | 42.00000 | 31.20000 | 24.31111 | 15.45397 | 6.53333 | 1.64444 | 11.25870 | 1.00000 |

**Table 8.** Judgment matrix (full packaging engineer A).

| Criteria | (C) | (Q) | (S) | (T) | (P) | (M) | Eigenvector | W (Weights) |
|---|---|---|---|---|---|---|---|---|
| (C) | 1.00000 | 0.20000 | 0.11111 | 5.00000 | 0.33333 | 0.33333 | 0.48075 | 0.04961 |
| (Q) | 5.00000 | 1.00000 | 0.11111 | 3.00000 | 5.00000 | 3.00000 | 1.70998 | 0.17648 |
| (S) | 9.00000 | 9.00000 | 1.00000 | 9.00000 | 7.00000 | 5.00000 | 5.42583 | 0.55996 |
| (T) | 0.20000 | 0.33333 | 0.11111 | 1.00000 | 0.33333 | 0.33333 | 0.30613 | 0.03159 |
| (P) | 3.00000 | 0.20000 | 0.14286 | 3.00000 | 1.00000 | 0.33333 | 0.66401 | 0.06853 |
| (M) | 3.00000 | 0.33333 | 0.20000 | 3.00000 | 3.00000 | 1.00000 | 1.10292 | 0.11383 |
| Σ (Sum) | 21.20000 | 11.06667 | 1.67619 | 24.00000 | 16.66667 | 10.00000 | 9.68962 | 1.00000 |

**Table 9.** Judgment matrix (full packaging engineer B).

| Criteria | (C) | (Q) | (S) | (T) | (P) | (M) | Eigenvector | W (Weights) |
|---|---|---|---|---|---|---|---|---|
| (C) | 1.00000 | 7.00000 | 9.00000 | 0.20000 | 0.20000 | 0.20000 | 0.89208 | 0.08426 |
| (Q) | 0.14286 | 1.00000 | 7.00000 | 0.11111 | 0.11111 | 0.14286 | 0.34759 | 0.03283 |
| (S) | 0.11111 | 0.14286 | 1.00000 | 0.11111 | 0.11111 | 0.11111 | 0.16710 | 0.01578 |
| (T) | 5.00000 | 9.00000 | 9.00000 | 1.00000 | 9.00000 | 5.00000 | 5.12993 | 0.48453 |
| (P) | 5.00000 | 9.00000 | 9.00000 | 0.11111 | 1.00000 | 0.20000 | 1.44225 | 0.13622 |
| (M) | 5.00000 | 7.00000 | 9.00000 | 0.20000 | 5.00000 | 1.00000 | 2.60847 | 0.24637 |
| Σ (Sum) | 16.25397 | 33.14286 | 44.00000 | 1.73333 | 15.42222 | 6.65397 | 10.58742 | 1.00000 |

**Table 10.** Judgment matrix (full packaging engineer C).

| Criteria | (C) | (Q) | (S) | (T) | (P) | (M) | Eigenvector | W (Weights) |
|---|---|---|---|---|---|---|---|---|
| (C) | 1.00000 | 0.20000 | 0.11111 | 5.00000 | 1.00000 | 5.00000 | 0.90668 | 0.08284 |
| (Q) | 5.00000 | 1.00000 | 0.11111 | 7.00000 | 7.00000 | 7.00000 | 2.39885 | 0.21916 |
| (S) | 9.00000 | 9.00000 | 1.00000 | 9.00000 | 9.00000 | 9.00000 | 6.24025 | 0.57012 |
| (T) | 0.20000 | 0.14286 | 0.11111 | 1.00000 | 0.20000 | 1.00000 | 0.29317 | 0.02678 |
| (P) | 1.00000 | 0.14286 | 0.11111 | 5.00000 | 1.00000 | 3.00000 | 0.78727 | 0.07193 |
| (M) | 0.20000 | 0.14286 | 0.11111 | 1.00000 | 0.33333 | 1.00000 | 0.31922 | 0.02916 |
| Σ (Sum) | 16.40000 | 10.62857 | 1.55556 | 28.00000 | 18.53333 | 26.00000 | 10.94545 | 1.00000 |

**Table 11.** Judgment matrix (full packaging engineer D).

| Criteria | (C) | (Q) | (S) | (T) | (P) | (M) | Eigenvector | W (Weights) |
|---|---|---|---|---|---|---|---|---|
| (C) | 1.00000 | 1.00000 | 0.11111 | 3.00000 | 1.00000 | 1.00000 | 0.83268 | 0.08253 |
| (Q) | 1.00000 | 1.00000 | 0.11111 | 7.00000 | 1.00000 | 1.00000 | 0.95898 | 0.09505 |
| (S) | 9.00000 | 9.00000 | 1.00000 | 9.00000 | 9.00000 | 9.00000 | 6.24025 | 0.61850 |
| (T) | 0.33333 | 0.14286 | 0.11111 | 1.00000 | 0.20000 | 0.20000 | 0.24412 | 0.02420 |
| (P) | 1.00000 | 1.00000 | 0.11111 | 5.00000 | 1.00000 | 1.00000 | 0.90668 | 0.08986 |
| (M) | 1.00000 | 1.00000 | 0.11111 | 5.00000 | 1.00000 | 1.00000 | 0.90668 | 0.08986 |
| Σ (Sum) | 13.33333 | 13.14286 | 1.55556 | 30.00000 | 13.20000 | 13.20000 | 10.08939 | 1.00000 |

**Table 12.** Judgment matrix (junior packaging engineer).

| Criteria | (C) | (Q) | (S) | (T) | (P) | (M) | Eigenvector | W (Weights) |
|---|---|---|---|---|---|---|---|---|
| (C) | 1.00000 | 0.33333 | 0.33333 | 5.00000 | 1.00000 | 0.33333 | 0.75498 | 0.09841 |
| (Q) | 3.00000 | 1.00000 | 1.00000 | 5.00000 | 1.00000 | 0.33333 | 1.30766 | 0.17044 |
| (S) | 3.00000 | 1.00000 | 1.00000 | 5.00000 | 1.00000 | 0.33333 | 1.30766 | 0.17044 |
| (T) | 0.20000 | 0.20000 | 0.20000 | 1.00000 | 0.11111 | 0.20000 | 0.23713 | 0.03091 |
| (P) | 1.00000 | 1.00000 | 1.00000 | 9.00000 | 1.00000 | 0.20000 | 1.10292 | 0.14376 |
| (M) | 3.00000 | 3.00000 | 3.00000 | 5.00000 | 5.00000 | 1.00000 | 2.96177 | 0.38604 |
| Σ (Sum) | 11.20000 | 6.53333 | 6.53333 | 30.00000 | 9.11111 | 2.40000 | 7.67211 | 1.00000 |

**Table 13.** Judgment matrix (full packaging analyst).

| Criteria | (C) | (Q) | (S) | (T) | (P) | (M) | Eigenvector | W (Weights) |
|---|---|---|---|---|---|---|---|---|
| (C) | 1.00000 | 3.00000 | 0.20000 | 5.00000 | 5.00000 | 1.00000 | 1.57042 | 0.19034 |
| (Q) | 0.33333 | 1.00000 | 0.33333 | 7.00000 | 0.33333 | 3.00000 | 0.95898 | 0.11623 |
| (S) | 5.00000 | 3.00000 | 1.00000 | 9.00000 | 5.00000 | 5.00000 | 3.87298 | 0.46941 |
| (T) | 0.20000 | 0.14286 | 0.11111 | 1.00000 | 0.20000 | 1.00000 | 0.29317 | 0.03553 |
| (P) | 0.20000 | 3.00000 | 0.20000 | 5.00000 | 1.00000 | 1.00000 | 0.91839 | 0.11131 |
| (M) | 1.00000 | 0.33333 | 0.20000 | 1.00000 | 1.00000 | 1.00000 | 0.63677 | 0.07718 |
| Σ (Sum) | 7.73333 | 10.47619 | 2.04444 | 28.00000 | 12.53333 | 12.00000 | 8.25071 | 1.00000 |

The calculation of the priority eigenvector of the matrix rows was performed via application of the simple geometric mean, which is described as $GM = \sqrt[x]{X_1 \times X_2 \times X_3 \times X_n}$.

As for the analysis of the Consistency Index (CI), it was calculated by the ratio: $CI = (\lambda_{max} - n)/(n-1)$, where $\lambda_{max}$ is the maximum eigenvalue and, complementarily, the calculation of the Consistency Ratio (CR) governed by the ratio: CR = CI/RI, also depended on the number of criteria (*n*) used as constructs in the construction of the problem solution. The Random Index (RI), proposed by Saaty [20], can be selected according to the number (*n*) in Table 14 below.

**Table 14.** Random Consistency Index—adapted from [28].

| *n* | 1 | 2 | 3 | 4 | 5 | 6 | 7 | 8 |
|---|---|---|---|---|---|---|---|---|
| **Random Index (RI)** | 0.00 | 0.00 | 0.58 | 0.98 | 1.12 | 1.24 | 1.32 | 1.41 |

The values found for the CR must respect the condition of being ≤0.10 to be considered sufficient in the light of the AHP method.

Thus, by applying the AHP method, the CR values, shown in Table 15 below, were obtained for the experts' judgment matrices.

**Table 15.** Consistency analysis of the individual matrices.

| Function/Position | $\lambda_{max}$ | Consistency Index (CI = μ) | Consistency Ratio (CR = CI/RI) | Status (CR ≤ 0.10000) |
|---|---|---|---|---|
| **SCM Director** | 7.98623 | 0.39725 | 0.32036 | inconsistency |
| **SCM Manager** | 7.25073 | 0.25015 | 0.20173 | inconsistency |
| **Packaging Eng. Coordinator** | 7.58549 | 0.31710 | 0.25572 | inconsistency |
| **Full Packaging Engineer A** | 6.98206 | 0.19641 | 0.15840 | inconsistency |
| **Full Packaging Engineer B** | 7.73218 | 0.34644 | 0.27938 | inconsistency |
| **Full Packaging Engineer C** | 7.41608 | 0.28322 | 0.22840 | inconsistency |
| **Full Packaging Engineer D** | 6.41001 | 0.08200 | 0.06613 | consistency |
| **Junior Packaging Engineer** | 6.49279 | 0.09856 | 0.07948 | consistency |
| **Full Packaging Analyst** | 6.96541 | 0.19308 | 0.15571 | inconsistency |

Number of criteria (*n*) = 6; Random Index (RI) = 1.24.

In summary, only two with consistent values and seven with inconsistent values were identified during the matrices' analysis. However, as the specialists' goal was to build a decision-making model that takes into account the different judgments and opinions of all the members of the packaging engineering team, without any distinction or value judgment, they chose to proceed to the answer's aggregation step under the AIJ approach, detailed below.

3.4.5. Aggregate Model Matrix (Aij)

The present work aimed at establishing integration between the opinions of packaging engineering specialists of the same company. The AHP-GDM method was applied in the AIJ (aggregating individual judgment) form in order to prioritize a consensus among all the group components, without distinction of position.

Thus, due to the respondents' answers having been provided utilizing a judgment scale, and in order to consider it more relevant to have a "normalized average" value when aggregating the values explained in the respondents' matrices, the simple geometric average was used to fill in the fields of the group judgment matrix. In other words, to obtain the final answer for each of the pairwise comparisons between the criteria, the simple geometric mean of the answers of the nine experts on the team was calculated, thus establishing a form of aggregation of individual judgments. This way, by applying the AHP-GDM method, it was possible to obtain the following aggregated matrix for the group (see Table 16 below).

Similar to the analysis performed in the previous step, the Consistency Ratio (CR) was checked for the condition CR ≤ 0.10 (see Table 17).

It was found that the value calculated for the Consistency Ratio (CR = 0.02753) respects the condition CR ≤ 0.10, as established by the AHP method—validating the analysis. Finally, the weights for the criteria, shown in Table 18 and Figure 4 below, were obtained.

**Table 16.** Aggregate matrix AIJ (respondent group).

| Criteria | (C) | (Q) | (S) | (T) | (P) | (M) | Eigenvector | W (Weights) |
|---|---|---|---|---|---|---|---|---|
| **(C)** | 1.00000 | 0.48904 | 0.29756 | 2.12270 | 0.55854 | 0.63706 | 0.69211 | 0.10330 |
| **(Q)** | 2.04481 | 1.00000 | 0.36938 | 2.33225 | 1.13086 | 1.31386 | 1.17393 | 0.17521 |
| **(S)** | 3.36063 | 2.70720 | 1.00000 | 3.08776 | 1.84273 | 1.57113 | 2.08150 | 0.31068 |
| **(T)** | 0.47109 | 0.42876 | 0.32385 | 1.00000 | 0.29158 | 0.41341 | 0.44614 | 0.06659 |
| **(P)** | 1.79035 | 0.88428 | 0.54267 | 3.42949 | 1.00000 | 0.56334 | 1.08812 | 0.16241 |
| **(M)** | 1.56970 | 0.76111 | 0.63648 | 2.41889 | 1.77511 | 1.00000 | 1.21800 | 0.18179 |
| **Σ (Sum)** | 10.23660 | 6.27041 | 3.16996 | 14.39111 | 6.59884 | 5.49881 | 6.69982 | 1.00000 |

**Table 17.** Consistency analysis of the aggregate matrix.

| Matrix | $\lambda_{max}$ | Consistency Index (CI = μ) | Consistency Ratio (CR = CI/RI) | Status (CR ≤ 0.10000) |
|---|---|---|---|---|
| **Aggregate Matrix (Group)** | 6.17071 | 0.03414 | 0.02753 | consistency |

Number of criteria (*n*) = 6; Random Index (RI) = 1.24.

**Table 18.** Criteria weights from the aggregated matrix.

| Results | Criteria | | | | | | Total |
| | (C) | (Q) | (S) | (T) | (P) | (M) | (Σ) |
|---|---|---|---|---|---|---|---|
| W (Pesos) | 0.10330 | 0.17521 | 0.31068 | 0.06659 | 0.16241 | 0.18179 | 1.00000 |
| % | 10.33% | 17.52% | 31.07% | 6.66% | 16.24% | 18.18% | 100.00% |

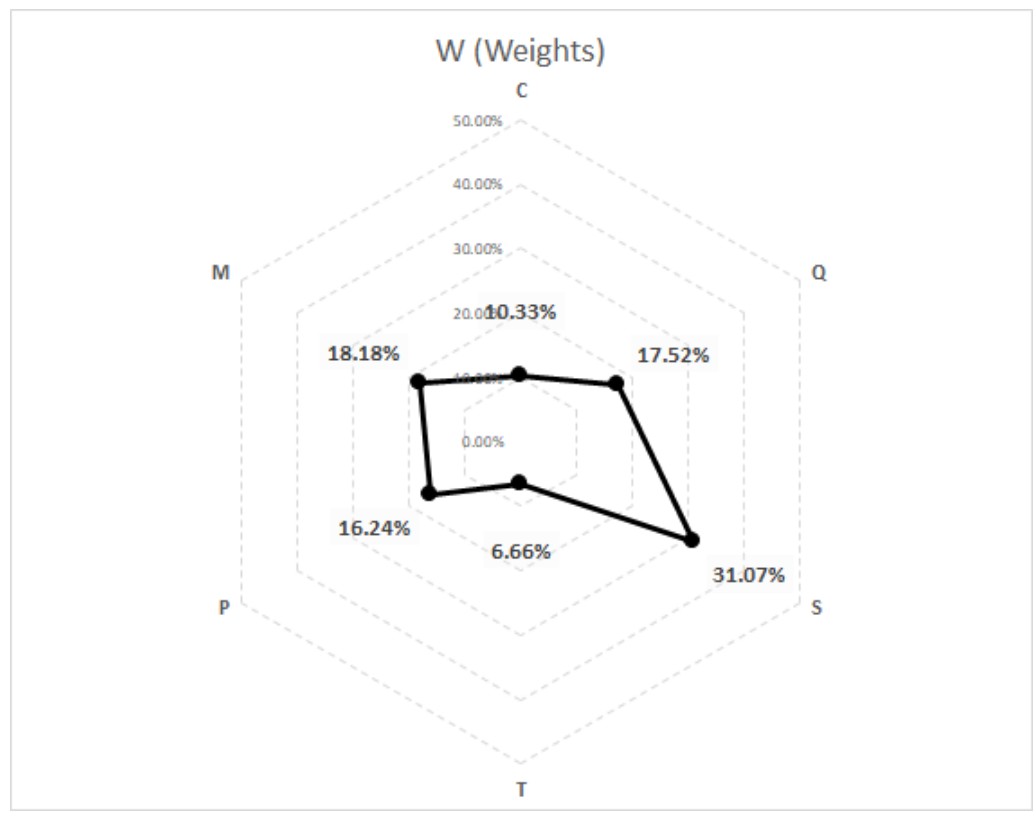

**Figure 4.** Radar distribution of the weights of the criteria in the aggregate matrix.

*3.5. Proposed Model*

With the weights assigned to each of the criteria, it was possible to create an indicator that will help experts in packaging development to establish individually (for each of the parts used by the assembler) its performance compared to other packaging definitions. Thus, the simplified equation that governs the model developed in this work is defined as follows:

$$\text{PPI}_i = 0.10330 \times (\text{C})_i + 0.17521 \times (\text{Q})_i + 0.31068 \times (\text{S})_i + 0.06659 \times (\text{T})_i + 0.16241 \times (\text{P})_i + 0.18179 \times (\text{M})_i$$

where:
PPI = Packaging Performance Indicator
C = Cost Criteria
Q = Quality Criteria
S = Safety Criteria
T = Time Criteria
D = Engineering Performance Criteria
G = Management Criteria
with $i$ = 1, 2, 3, ... (being $i$, each of the parts used by the automaker).

Once the suggested equation to quantify the individual performance of the set (part/ packaging) is defined, it is up to the specialists to align and define with their managers which subcriteria are inherent in each of the six macrocriteria that will be used to calculate the indicator.

## 4. Discussion and Practical and Theoretical Implications

The model proposed in this work aims to be a form of quick application, considering the restricted time available to the specialists in the area in their daily routines. With this in mind, those interested in applying the steps performed in this study do not need to have software and/or complex procedures at their disposal, requiring only the group's engagement. Additionally, the CEPA framework provides a simple way to guide the composite indicator composition process.

We can conclude that the AHP-GDM method is an alternative solution to the problem at hand through this study. This method, when combined with other tools and methods often used in the organizational environment, as is the case of brainstorming and data collection techniques via survey, proved to be valuable and capable of simplifying the performance analysis of the packaging used in the transit of parts between suppliers and the assembler, being an optional tool to help the decision-making process for the packaging development team.

During the model building process, specifically in the criteria selection stage, the specialists' know-how allowed them to prioritize the six most important criteria according to their experiences and perceptions, which enriched and increased the importance of establishing consensus/alignment. However, to broaden the discussion about which would be the essential criteria to be selected in new models or in future revisions, it is suggested to increase the sample of specialists to best contribute with their expertise. Finally, the study was limited to not performing a sensitivity analysis based on the weights assigned by each of the nine respondents. This does not preclude its analysis.

## 5. Conclusions

Since it is a model of simple application, it can be used, for example, in the elaboration of a ranking of which packagings, according to the criteria established by the specialists, need to be improved based on comparisons made between the other packagings listed, favoring the identification of which items should be improved as a priority. For this, a normalization of the values found in each of the scored subcriteria is enough. Despite being an applicable model for this case, the solution aims to alert the areas of logistics and processes to look for opportunities to implement improvement proposals in their areas of activity, especially concerning sustainability, in which case aspects related to sustainable

production and consumption and other CE practices could be further explicated and analyzed instead of being implicit in other criteria.

Having presented all these suggestions of possible indicators that could be implemented, we also see the opportunity, after a preliminary ranking (which classifies the items on a combined scale of the ABC/XYZ type—separated by color), to illustrate the values found graphically and intuitively, allowing those involved to identify which items should be prioritized quickly. Thus, new approaches can be structured according to the interests of specialists and the organization itself, based on the discussed model, including the addition of criteria and/or parameters related to sustainability and circular economy or the combination with other methods, such as: fuzzy, TOPSIS, or ARAS.

## 6. Suggestions for Future Work

In addition to the suggestions made throughout the text, the model built in this work can also be implemented in other areas and/or manufacturing processes that involve industrial packaging, such as in the case of manufacturing companies: electronics, chemicals, cosmetics, food products, among others. Another opportunity for application is the possibility of correlating this study with the principles and requirements established in the NBR ABNT ISO 55.0001:2014 standard, referring to asset management. Since the industrial equipment owned by organizations is also considered an asset, it has economic value, and the proper functioning of this equipment reduces the risks inherent in the manufacturing process. Finally, after the suggestions pointed out for further deepening future research, we highlight the possibility and the great potential for the development of integrated software and/or an application that can be used as a tool for the packaging engineering specialists in their daily work.

**Author Contributions:** Conceptualization, M.M.d.C. and R.G.G.C.; data curation, M.M.d.C. and R.G.G.C.; formal analysis, M.M.d.C., R.G.G.C. and R.S.S.; funding acquisition, R.G.G.C.; investigation, M.M.d.C.; methodology, M.M.d.C., R.G.G.C. and R.S.S.; project administration, R.G.G.C.; resources, M.M.d.C.; software, M.M.d.C., R.G.G.C. and R.S.S.; supervision, R.G.G.C.; validation, R.G.G.C.; visualization, M.M.d.C. and R.S.S.; writing—original draft, M.M.d.C.; writing—review and editing, R.G.G.C. and R.S.S. All authors have read and agreed to the published version of the manuscript.

**Funding:** This work was supported by the Coordenação de Aperfeiçoamento de Pessoal de Nível Superior—Brazil—CAPES (Finance Code 001), and the Carlos Chagas Filho Foundation for Research Support of the State of Rio de Janeiro—FAPERJ (grant number E-26/201.363/2021; E26/211.298/2021).

**Institutional Review Board Statement:** Not applicable.

**Informed Consent Statement:** Not applicable.

**Data Availability Statement:** Not applicable.

**Conflicts of Interest:** The authors declare no conflict of interest.

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
