# Peer review of "Industrial Packaging Performance Indicator Using a Group Multicriteria Approach: An Automaker Reverse Operations Case"

_logistics, 2022_

Round 1

Reviewer 1 Report

A literature review is scarce regarding the papers which deal with AHP-GDM method. The authors state "However, there are still few studies related to this method…." so they should expand subsection 2.4 AHP-GDM with the articles related to this method with special emphasis on articles related to the area of packaging development.

The key questions that the authors have to answer are:

- What makes this paper original (scientific) taking into account that AHP and group decision making are already known methods in the literature.

- Is the proposed model (PPI) better than the standard AHP procedure taking into account that we can use software to quickly calculate the optimal solution via the standard (full procedure) of the AHP method?

Reviewer 2 Report

I am very grateful for having been able to participate in the review of this paper on certain very interesting aspects of the circular economy and reverse logistics that I believe may be crucial for the future of a more sustainable society.

The introduction is clear and concise, the literature review is good and perhaps I miss first-rate and closely related articles such as https://doi.org/10.3390/su11164380, which would be very interesting and valuable to see reflected in the paper. The methodology is clear, and the results and conclusions well founded and well developed. I think that by adding some current and valuable references in the Literature Review section this paper is a very good candidate for publication.
